# Effectiveness of On-Court Resistive Warm-Ups on Change of Direction Speed and Smash Velocity during a Simulated Badminton Match Play in Well-Trained Players

**DOI:** 10.3390/jfmk6040081

**Published:** 2021-09-27

**Authors:** Man Tong Chua, Kin Ming Chow, Danny Lum, Andrew Wei Han Tay, Wan Xiu Goh, Mohammed Ihsan, Abdul Rashid Aziz

**Affiliations:** 1Sport Science & Sport Medicine, Singapore Sport Institute, Sport Singapore, Singapore 397630, Singapore; nie21.cmt@e.ntu.edu.sg (M.T.C.); chow_kin_ming@sport.gov.sg (K.M.C.); danny_lum@sport.gov.sg (D.L.); andrew_tay@sport.gov.sg (A.W.H.T.); goh_wan_xiu@sport.gov.sg (W.X.G.); 2Physical Education and Sports Science Academic Group, National Institute of Education, Nanyang Technological University, Singapore 639798, Singapore; 3Human Potential Translational Research Program, Yong Loo Lin School of Medicine, National University of Singapore, Singapore 119077, Singapore; ihsan.m@nus.edu.sg

**Keywords:** muscle potentiation, wearable resistance, dynamic, resistant exercise, agility, quickness

## Abstract

In badminton, power production can be enhanced through the fundamental practice of a dynamic warm-up with resistance conditioning activity to induce a post-activation performance enhancement (PAPE) effect. The use of heavy resistance exercise in the form of heavy weights to induce PAPE during competition is not logistically practical in the badminton arena. Thus, there is a need to investigate the use of easily available alternative preconditioning stimuli to induce a similar potentiating effect in badminton-specific performance. This study adopted a repeated-measures design of three warm-up conditions: control (CON), weighted wearable resistance (WWR), and resistance band variable resistance (BVR). Fourteen badminton players from the national training squad (11 males, 3 females, age 18 ± 1 y) completed the experimental sessions in random order. Change of direction speed (CODS) and smash velocity (SV) tests were performed at five timepoints—baseline test after the warm-up and at the end of each of the four exercise blocks of a simulated match play protocol. CODS was significantly faster under the two resistance warm-up conditions (WWR and BVR) compared to the CON condition at baseline (−0.2 s ± 0.39 and −0.2 s ± 0.46, *p* = 0.001 and 0.03, *g* = 0.47 and 0.40, respectively), but there were no differences at the other timepoints (all *p* > 0.05). SV was significantly faster for all the four exercise blocks than at baseline under all three warm-up conditions (*p* = 0.02), but there were no differences in SV between the three warm-up conditions across all the five measured timepoints (*p* = 0.15). In conclusion, implementing resistance (~10% body weight) in sport-specific plyometric exercises using WWR or BVR during warm-up routines may induce PAPE effects on the change of direction speed but not smash velocity, in well-trained badminton players, as compared with the same warm-up exercises using bodyweight (i.e., CON condition). The positive effects of CODS were, however, observed only at the start of the match and possibly lasted for up to between 5 and 10 min of match play.

## 1. Introduction

Badminton is characterized as a ballistic intermittent sport, with match durations lasting up to 40 min to 1 h, and a temporal structure distinguished by repeated high-intensity, short-duration efforts [1]. Physical components, including change of direction speed (CODS) and smash velocity (SV), are critical to positive match outcomes. CODS is the ability to rapidly change direction; and in badminton, the player engages in various multidirectional movement patterns in rapid succession to make defensive/counterattacking retrievals or create attacking plays [2,3,4]. Likewise, the ability to execute a powerful overhead smash is an important determinant of success in both the men’s and women’s games, especially as a finishing shot in majority of rallies [1]. Indeed, the winning factor for a match is often attributed to high SV during smash execution [5].

The fundamental practice of dynamic warm-ups via various priming strategies is often employed to enable optimal physical performance during match play [6,7]. In recent years, the concept of post-activation performance enhancement (PAPE) has been included as part of warm-up routines to achieve maximal motor unit and neural recruitment before the criterion performance [8]. PAPE is a phenomenon characterized by improved muscular power after a “conditioning” contractile activity [9,10,11]. The use of heavy resistance exercise is a popular method to elicit PAPE prior to intermittent sport performance [8]. Among football players, Zois and colleagues (2011) [12] observed that a 5-repetition maximum (5RM) leg press warm-up was more effective in improving reactive agility, countermovement jump, and 20 m sprint performance compared with traditional warm-up and small-sided games. However, the application of heavy resistance exercises, especially with the use of weights or gym equipment, is not logistically feasible in badminton competition venues. Hence, it is essential to investigate alternative methods that can be easily implemented in the field, and able to effectively provide preconditioning stimuli to induce a potentiating effect.

The use of wearable resistance (weighted vest or wearable limb weights) or resistance bands may be robust, feasible alternatives that can provide sufficient intensity to enhance the potentiating effect of a warm-up [13,14,15]. The performance of dynamic warm-up activities with the use of a weighted vest has been shown to increase lower-limb power output and countermovement jump performance in female athletes [16] and CODS in professional badminton players [15]. Likewise, the implementation of resistance bands during warm-ups has been shown to improve the performance of the Special Judo Fitness Test and peak power of the barbell high pull [14], as well as roundhouse kick velocity in Taekwondo athletes [13]. However, these studies only investigated the immediate effects of resisted warm-ups on short bouts of performance tests. It is unknown how such beneficial effects translate to key physical components during badminton match play.

In a prolonged intermittent racket sport such as badminton, the qualities of CODS and SV within the duration of the match are key physical factors that may ultimately determine the match outcome (i.e., win or lose). Therefore, the aim of this study was to investigate the effects of portable resistance modalities, i.e., weighted wearables (vest and wrist weight) and resistance bands during warm-up on CODS and SV performance prior to and during a simulated match play amongst trained badminton players. It is hypothesized that including resistance modalities in a badminton warm-up routine would positively enhance CODS and SV performance just before and throughout a simulated match play.

## 2. Materials and Methods

### 2.1. Participants

Fourteen healthy, trained badminton players in the national training squad, 11 males (age 19 ± 2.6 y, stature 177 ± 7.3 cm, body mass 69 ± 6.8 kg, playing category: 3 singles, 8 doubles) and 3 females (age 19 ± 1.2 y, stature 168 ± 8.7 cm, body mass 62 ± 3.1 kg, playing category: 3 doubles), were recruited. Stringent inclusion criteria were adhered to in the recruitment of participants as performance tests required high skill levels—players must be nationally ranked (top 8) either in the singles’ or doubles’ categories, with at least two years of experience with full-time strength and conditioning training (minimally two sessions a week). All players have competed at regional and international competitions. All players were briefed of the risks and benefits of the study and provided written informed consent prior to the commencement of the study. Consent was provided by parents/guardians for players < 21 y old. Ethics approval was obtained from the Institutional Review Board.

### 2.2. Experimental Approach

The study adopted a repeated-measures design of three experimental warm-up conditions. All trials were randomized, cross-over, and counterbalanced across all players. Players underwent one familiarization session and three experimental sessions separated between two to seven days apart. The study was part of the players’ training program during the off-season period. All familiarization and experimental measurements were led and conducted by a primary investigator for consistency in data collection procedures.

During familiarization, players were briefed on the procedures of the study and were informed that the aim of the study was to determine the ‘ideal’ or optimal warm-up for badminton players. This was to blind them from the true purpose of the study—which was to determine the effect of PAPE via two types of resistant warm-up strategies on CODS and SV performance in a match play setting relative to the typical badminton pre-match warm-up. Following the briefing, players underwent a body composition assessment using a body composition analyzer (Inbody 770, Seoul, Korea). The players then proceeded to familiarize with the various exercises of the three warm-up conditions. The determination of the load and type of resistance bands used was also trialed during this session. Lastly, players performed a condensed version of the experimental trial—they completed a baseline CODS and SV test, followed by one exercise block of the simulated match play protocol (SMP), which consisted of 5 min of simulated rallies followed by CODS and SV tests. Players recorded their food and fluid intake 24 h before their first trial and were instructed to replicate the same diet (as close as possible) 24 h prior to all subsequent trials.

The experimental trials (Figure 1) were conducted in an indoor hall where the players routinely train. Upon arrival, players performed their randomly allocated warm-up condition. Four minutes after the warm-up, the players performed the baseline CODS and SV performance tests. Subsequently, the players rested for one minute and began to perform the SMP. The SMP, excluding the warm-up, lasted for a total duration of ~35 min, which is approximately the duration of one set of competitive match play (depending on category) [1].

### 2.3. Warm-Up Protocols

All three warm-up protocols consisted of similar dynamic stretches, sport-specific movements, and plyometric exercises–the primary difference was the use of external resistance specifically during the performance of sport-specific and plyometric exercises (Table 1). Under the control (CON) condition, players used their own bodyweight (no external resistance), while in the weighted wearable resistance (WWR) and banded variable resistance (BVR) groups, weighted vest + wrist weights and resistance bands were used, respectively. Exercise nos. 1 to 19 were performed at the player’s own pace under all three warm-up conditions, with no more than 15 s of passive recovery between exercises. Following that, depending on the condition, the player performed sport-specific plyometric exercise nos. 20 to 24 either with WWR, BVR, or without external resistance (CON). These exercises were performed with maximal effort, and with 1 min of passive recovery between each set of exercise.

In WWR, a weighted vest (Body Sculpture, Bradford, Yorkshire, UK) and wrist weight (Exogen Forearm Sleeves, Lila Movement Technology, Kuala Lumpur, Malaysia) were loaded to approximately ~10% (up to ~12% due to unavailability of certain weights) of the player’s body mass and segmental limb (playing arm) mass [15]. The mass of the players’ playing arm (~3.1 ± 0.5 kg) was estimated during the body composition analysis. Under the BVR condition, the resistance level of resistance bands (Sanctband, Sanctuary Health, Perak, Malaysia) utilized for the push-ups and overhead smash was the heaviest band, which allowed for a full range of motion [14]. For banded bilateral countermovement jumps, alternating split squat jumps, and 4-corner shadow movement, a band with a resistance of approximately 10% of the player’s body weight (BW) was employed [17]. Different bands for each player were tested for suitability during the player’s familiarization trial.

### 2.4. Simulated Match Play

The SMP consisted of 4 exercise blocks (Figure 1). Each exercise block included 5 min of simulated rallies, followed by CODS and SV performance tests. The 5 min of simulated rallies incorporated 15 rallies of 5 shots where shuttlecocks were fed by a feeder from the opposite side of the court to the various zones (Figure 2): Zone 1 (Z1), Zone 2 (Z2), Zone 3 (Z3), and Zone 4 (Z4). Each 1.5 m × 1.5 m zone was demarcated by small tape markers on the court. For each rally, 5 shuttlecocks were fed to players at a 2 s interval (dictated by beeping sounds) in a random order until a total number of 3 feeds to Z3/Z4 and a total number of 2 feeds to Z1/Z2 were achieved. Players returned with a net shot or underhand clear shot from the feeding shots in Z1 and Z2, and with either a drive or overhead clear shot from Z3 and Z4. Players were provided with a 30 s rest at the end of the 5 min simulated rallies to prepare for the CODS and SV tests, and 1 min of passive rest after the CODS and SV tests before proceeding to the next exercise block. Heart rate ((HR); H10, Polar Electro Oy, Kempele, Finland) was monitored throughout the experimental session while ratings of perceived exertion (6–20 RPE scale, Borg, 1982; reported in arbitrary unit, au) were administered after warm-up and after each exercise block.

### 2.5. Change of Direction Speed and Smash Velocity

The CODS test (Figure 2) was adapted from a previously published badminton-specific multiple repeated-ability test [18], where the intra-class correlation coefficient (ICC) and coefficient of variation (CV) of the mean CODS time of elite players were reported to be 0.95 and 3.9%, respectively. Throughout the CODS and SV tests, players were constantly reminded not to pace themselves, but to perform every CODS sprint and badminton smash with maximal effort. At the start of the test, players assumed the ready position at the center zone (CZ) of the court, and 20 cm away from the starting line of the CZ, which would trigger the light gates (Speed Light Sports Timing System, Swift Performance Equipment, Lismore, NSW, Australia). In the test, players then moved to all four corners of the court from the CZ using badminton-specific footwork and were required to lunge and touch the cones with the racket in the following sequence: C1 (left forecourt)-CZ-C2 (right forecourt)-CZ-C3 (right backcourt)-CZ-C4 (left backcourt)-CZ, before sprinting to the back of the court to complete the test. For each CODS test, players had to complete 3 sets of the same pattern, with a 1:2 work:rest ratio. The performance measure was the mean time of the 3 CODS trials assessed at each timepoint (i.e., after warm-up, and at the end of each exercise blocks 1, 2, 3, and 4).

The SV test was performed ~10 s after completing the CODS test (Figure 2). A square target of 1.5 m × 1.5 m was marked at the upper forehand side of the badminton court. This allowed the player to execute their smash stroke at the highest velocity while maintaining a good level of precision. A radar gun (Stalker ATS II Radar Gun, Applied Concepts, TX, USA) was positioned 3 m behind the player strike area and behind the badminton court, aligned to the approximate height of the shuttlecock smash release, approximately 2.5 m and in the direction of the target. This setup was akin to that of Phomsoupha and Laffaye [19], who reported an ICC of 0.96 and a CV < 4.3%. The players then performed nine forehand smashes at maximum power onto the target area. These smashes were executed from the lob feeds by the investigator from the other side of the court, delivered within 5 s intervals between each feed. The same investigator performed the feedings for all players throughout the study. The performance measure was determined by the average velocity of the two fastest smashes during each exercise block (this criterion was, however, not mentioned to the players, to ensure that players put in their maximal effort during all the smashes throughout the session).

### 2.6. Statistical Analysis

Data were analyzed using statistical software (IBM SPSS Version 24, Chicago, IL, USA). One-way analysis of variance (ANOVA) was conducted for measurements of mean HR, RPE, CODS, and SV between the three warm-up conditions. Two-factor trial × time repeated measures of ANOVA was conducted for measurements of CODS, SV, RPE, and HR under the three warm-up conditions across all five timepoints, as well as measurements of CODS and SV at baseline vs. mean of CODS and SV in all four exercise blocks between the three warm-up conditions. Mauchly’s test was consulted, and either Green–Geisser or Huynh–Feldt correction was applied if sphericity was violated. If a significant main effect was observed, post hoc *t*-tests with Bonferroni adjustments were used to detect the occurrences of the differences. The significance level was set at *p* < 0.05. Effect sizes of main effects were computed via partial eta-squared (η2p) and was deemed to be without effect if 0 < η2p ≤ 0.01; small if 0.01 < η2p ≤ 0.06; moderate if 0.06 < η2p ≤ 0.14; and large if η2p > 0.14 [20]. For pairwise comparisons, effect sizes were calculated using Hedges g (*g*) and is deemed to be small in effect if *g* ≥ 0.2; medium if *g* ≥ 0.5; and large if *g* ≥ 0.8 [21]. The smallest worthwhile change (SWC) of CODS across all timepoints in CON was calculated by the equation: standard deviation (SD) × 0.2 [22]. To examine if CODS under WWR and BVR conditions did elicit the SWC at each of the five timepoints, the percentage difference (%DIFF) of CODS in WWR or BVR from CODS–SWC in CON was calculated using the equation: ((CODS in WWR or BVR)-(CODS-SWC in CON))/(CODS-SWC in CON) × 100%, with %DIFF ≤ 0% deemed to elicit SWC and %DIFF ≥ 0% deemed to not elicit any SWC.

## 3. Results

There were no significant interaction effects of condition × time for both HR and RPE across warm-up and exercise blocks 1 to 4 (*p* = 0.55 and 0.55, η2p = 0.05 and 0.06, respectively). However, there were significant main effects for time for both HR and RPE (both *p* < 0.01, η2p = 0.96 and 0.73, respectively), with both variables increasing progressively throughout the duration of exercise. Importantly, no significant differences were found between the overall session HR (CON: 139 ± 13 b·min^−1^, WWR: 141 ± 11 b·min^−1^, and BVR: 138 ± 10 b·min^−1^) and RPE (CON: 6.0 ± 1.4 au, WWR: 6.0 ± 1.6 au, and BVR: 6.0 ± 1.3 au) between all three warm-up conditions (*p* = 0.82 and 0.99, η2p = 0.01 and 0.001, respectively). These data indicated that all three warm-up protocols elicited similar physiological stresses and perceptual responses from the participants during the SMP.

Table 2 depicts the descriptive statistics of CODS and SV at baseline, exercise blocks 1 to 4, mean CODS of the four exercise blocks, and mean CODS of all five timepoints measured during the session, under the three warm-up conditions. No significant differences were observed between the mean CODS of all five timepoints in the three warm-up conditions (*p* = 0.29, η2p = 0.01). There were no significant interaction effects of condition × time for CODS between the three warm-up conditions at all five timepoints. (*p* = 0.09, η2p = 0.14; Figure 3a). However, there were significant main effects of time under all three warm-up conditions (*p* = 0.001, η2p = 0.38), where CODS was significantly faster in exercise block 4 than at baseline (~1.63 s, *p* = 0.04, *g* = 0.39) and exercise block 1 (~1.05 s, *p* = 0.03, *g* = 0.26) under CON, WWR, and BVR conditions (Figure 3a). There were significant interaction effects of condition × time for CODS when comparing between baseline CODS values to the mean of the four exercise blocks (*p* = 0.02, η2p = 0.27; Figure 4a). Post hoc tests revealed a significantly faster CODS at baseline under WWR and BVR conditions as compared to the CON condition (both ~0.2 s, *p* = 0.001 and 0.03, *g* = 0.47 and 0.40, respectively). The %DIFF in CODS of WWR and BVR from CODS–SWC in CON at all five timepoints are depicted in Figure 5. The SWC in CODS was observed in WWR (Figure 5a) at baseline (−1.53%), exercise block 1 (−0.35%), exercise block 2 (−0.53%), and exercise block 3 (−0.27%), but not in exercise block 4 (+0.01%). The SWC in CODS was achieved in BVR (Figure 5b) only at baseline (−1.48%).

There were no significant differences observed between the mean SV of all five timepoints under the three warm-up conditions (*p* = 0.96, η2p = 0.00). Furthermore, no significant interaction effects of condition × time for SV were observed when comparing the three conditions across all five timepoints, from baseline to exercise block 4 (*p* = 0.15, η2p = 0.12; Figure 3b). There were also no significant interaction effects of condition × time for SV when comparing baseline SV values to the mean of all the four exercise blocks (*p* = 0.40, η2p = 0.07; Figure 4b). However, there were significant main effects of time factor under all three warm-up conditions (*p* = 0.02, η2p = 0.34, Figure 4b). SV was significantly faster in the exercise blocks than during baseline under all three warm-up conditions (~2.6 m·s^−1^), *p* = 0.02, *g* = 0.30).

## 4. Discussion

This study compared the effects of three different warm-up routines: (i) control, (ii) weighted wearable resistance, and (iii) banded variable resistance, on the change of direction speed (CODS) and smash velocity (SV) during a simulated match play protocol in trained badminton players. The study’s main findings were: (i) CODS was observed to be faster at baseline (4 min after warm-up) in both WWR and BVR compared with CON, but the improvements in CODS were more pronounced in WWR than BVR as exercise duration increased (exercise blocks 1–4); (ii) no difference in SV at all time points between the three warm up conditions.

The significantly faster CODS (−0.2 s, −2.7%) observed at baseline in WWR as compared with CON is in agreement with Maloney et al. (2014) [15] and Turki et al. (2020) [23]. In Maloney et al. (2014) [15], the 5–10% BW loaded weighted vest included during badminton-specific plyometric exercises in the warm-up routine improved CODS in trained badminton players up to the 6 min mark, while in Turki et al. (2020) [23], a 5–15% BW-loaded weighted vest in the final three soccer-specific exercises improved the repeated CODS performance in young soccer players regardless of recovery times (i.e., 15 s, 4 min, or 8 min) after warm-up [22]. Maloney et al. (2014) [15] further postulated that the loaded warm-up might have induced an acute increase in lower limb stiffness, enabling a higher force contribution from the stretch-shortening cycle, which was a pivotal element of an improved CODS [16,24].

With regards to BVR condition, this study adopted Lum and Chen’s (2020) [17] recommendation of a loading variable resistance intensity of 10% BW to the lower body plyometric exercise during warm-ups to induce an optimal PAPE effect—which was previously only exemplified by improvements in countermovement jump (CMJ) qualities (i.e., CMJ height, time to peak force, etc.). This is the first study to observe that the banded variable resistive plyometric exercise during warm-up can also produce a PAPE effect to improve ballistic movement such as CODS acutely, which was observed to be significantly faster (−0.2 s or −2.7%) under the BVR than CON condition at baseline. Furthermore, past studies that investigated banded variable resistive warm-ups were only conducted in the sports of martial arts such as—Taekwondo [13] and Judo [14]. Lum (2019) [14] demonstrated that supplementing judo athletes’ warm-up routine with a resistance band pull (a judo-specific movement) and standing broad jump at maximal effort improved power output and performance in the special judo fitness test. Similarly, Aandahl et al. (2018) [13] reported a 3.3% improvement in roundhouse kick performance alongside higher rectus femoris EMG activity, when added resistance was supplied by elastic tubes attached to the ankle during maximal roundhouse kicks warm-ups. The authors suggested that the enhanced performances could be attributed to the increased recruitment of motor units, greater synchronization of motor units involved, and a reduction in pre-synaptic inhibition [13]. Similar mechanisms could explain the faster badminton-specific CODS after the BVR warm-up routine in the present study.

The ergogenic benefits observed in CODS after WWR and BVR conditions in the initial stages of a simulated match protocol have important match implications. Barreira and colleagues (2016) [25] reported that during the 2015 World Badminton Championships, 78% of players who led the scoreboard for the first seven points of a set would go on to win the set. This highlights the importance of the players to be at their optimal physical state right from the first rally of the match. Baseline values of CODS under WWR and BVR conditions were approximately 0.2 s (or 2.7%) faster compared with the CON condition. Considering the heavy temporal demands of the badminton match play, i.e., shot frequency of ~0.9–1.3 shots per second [1], being physically quicker at the beginning of the match (i.e., first 5–10 min) would be advantageous in allowing players to respond quicker to shots when making defending and attacking plays during the rallies. However, although both WWR and BVR elicited significantly faster CODS than CON at baseline, only the WWR warm-up condition seemed to maintain some of the ergogenic (albeit not statistically significant) benefits of improved CODS relative to CON as exercise or match duration increased. This was observed from baseline to exercise block 3 where CODS in WWR was faster than CODS–SWC in CON (Figure 5). Differences in the type of external load imposed on the player, i.e., consistent load of WWR vs. variable load of BVR conditions, may have affected potential factors influencing the extent of the PAPE effect, i.e., increases in muscle temperature and muscle water, and enhancements in muscle activation/neural drive and motivation/arousal levels [11]. It would be of interest to conduct further investigations into the differences in PAPE mechanisms between the WWR and BVR modalities in improving CODS for an extended duration of time.

During the smash action, force is being transferred from the lower limbs to the upper limbs through the kinetic chain [26], and as a higher vertical ground reaction force is moderately correlated (*r* = 0.6) to high shuttle velocity [27], the present investigators postulated that implementing resistance to sport-specific plyometric upper and lower movements would benefit smash velocity through the PAPE effect. However, the findings indicated that SV was not affected by the two resistant warm-ups compared to the CON condition across all five timepoints (Figure 3b). Interestingly, it was observed that SV was significantly higher during the exercise blocks than at baseline under all three conditions (Figure 4b), even though players were experiencing greater physical load as the match simulation progressed. The smash is a complex technical skill which involves more than just components of muscular strength and power [1,19]. Some of these components include visual reaction time (reaction to stimulus), body coordination (coordination of many body parts to a stimulus) [28], and stroke production (smash technique—backswing, shuttle contact point, and follow-through) [29]. Hence, the PAPE effects that the resisted warm-ups could have induced in improving muscular power would likely not be as pronounced. Therefore, it may be additionally important to include external stimuli, i.e., hitting the shuttlecock, when performing warm-up for a highly technical, coordinated skill such as the smash, in addition to solely performing the explosive plyometric exercises.

The strength of the current study lies with the incorporation of simulated match play to increase the ecological validity of this study. Previously, Maloney et al. (2014) [15] investigated the effectiveness of weighted vest loaded warm-up in badminton players only by comparing baseline and post-intervention (at 0, 2, 4, and 6 min after warm-up) vertical jump and CODS values [15]. The inclusion of the SMP into the present investigation allowed investigators to observe the effects of PAPE throughout an extended exercise duration—CODS and SV performance tests did not show any decline throughout the four exercise blocks, but were improved. The CODS in exercise block 4 was 0.11 s (1.5%) and 0.2 s (2.7%) faster than in exercise block 1 and baseline, respectively (Figure 3a), while SV was 2.6 m·s^−1^ (4.3%) faster in the exercise blocks as compared to baseline (Figure 4b). This is synonymous with past studies that observed no appreciable decrease in muscular performance (countermovement jump height, force production, repeated agility tests, etc.) in trained badminton players pre- and post-badminton match play [30,31,32]. This is likely due to the nature of the tests, which are short, explosive, and highly reliant on the immediate phosphocreatine energy system, which depletes albeit quickly, but is also rapidly re-synthesized during recovery [31,33]. In relation to the improvements of CODS and SV during the exercise blocks, investigators postulate that this could be due to the additional PAPE effect caused by the rapid, explosive movements when performing the SMP and the performance tests that are embedded between warm-up and exercise blocks. This was similar to the investigation of Abián et al. (2015) [31], where the authors did observe that, although not significant, measurements of footwork test, maximal isometric test, and agility *t*-test conducted on badminton players were improved by ~0.4%, ~2.9%, and 1.1%, respectively, after a 45 min simulated badminton match. Nonetheless, as muscular fatigue is unlikely to set in during a badminton match, the main window of opportunity for improved physical performance is therefore at the start and during the early stage of match play. Future studies may investigate the effects of resistive warm-ups on physical performance variables throughout a longer timeframe, i.e., throughout a tournament, where matches are played daily, and hence physical fatigue will inherently manifest in the later matches. A limitation observed under the BVR condition was the resistance bands used during warm-up for upper body explosive exercises–they were the heaviest that the participant could perform with a full ROM, as recommended in Lum (2019) [14]. In this study, investigators observed that when the heavy banded push-ups and overhead smash were performed, the concentric phase in these movements took a longer time compared with the lower body plyometric exercises performed with a resistance band of ~10% BW. Explosive movements that require a longer concentric phase than brief stretch shortening cycle movements such as the smash may limit the ability of the muscles to produce the rapid force needed during fast contraction velocities [34]. Moreover, heavier bands could have caused greater neuromuscular fatigue due to increased loading [17]. Therefore, it would be worthwhile to investigate into using lighter resistance bands to possibly induce the PAPE effect on badminton smash.

## 5. Conclusions

Implementing resistance (~10% BW) in sport-specific plyometric exercises via WWR or BVR during warm-up routines may induce optimal PAPE effects on the change of direction speed, but not smash velocity, in well-trained badminton players at the start of match play and, perhaps, up to 5–10 min into the match, as compared to the same warm-up exercises using bodyweight (i.e., CON condition).

## 6. Practical Implications

Given that CODS is a critical performance determinant in most intermittent sports [35,36], athletes can gain a competitive edge by including a resistance of ~10% BW to their sport-specific plyometric warm-up exercises to enhance change of direction speed at the commencement of their sporting event/match. This can be achieved with weighted wearables or resistance bands, which provide a portable and feasible alternative to traditional weights that may not be available at international competition venues.

## Figures and Tables

**Figure 1 jfmk-06-00081-f001:**
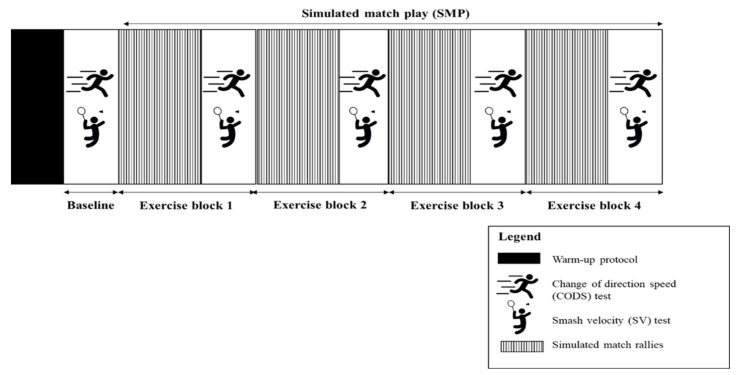
Study experimental procedure.

**Figure 2 jfmk-06-00081-f002:**
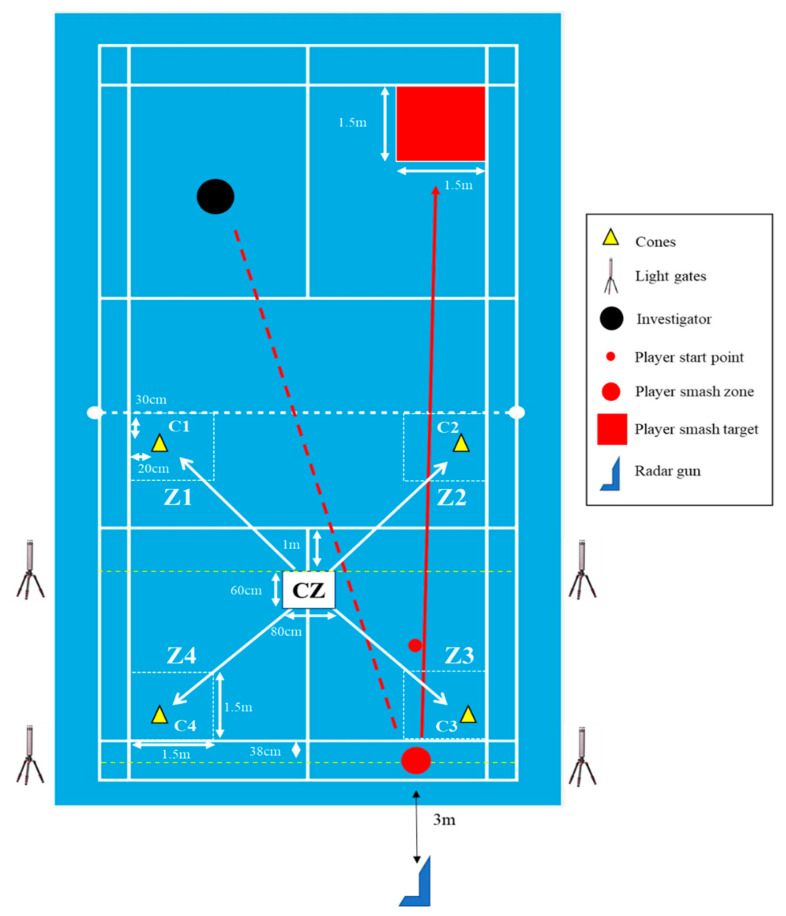
Layout of badminton court for simulated match rallies, change of direction speed (CODS), and smash velocity (SV) tests.

**Figure 3 jfmk-06-00081-f003:**
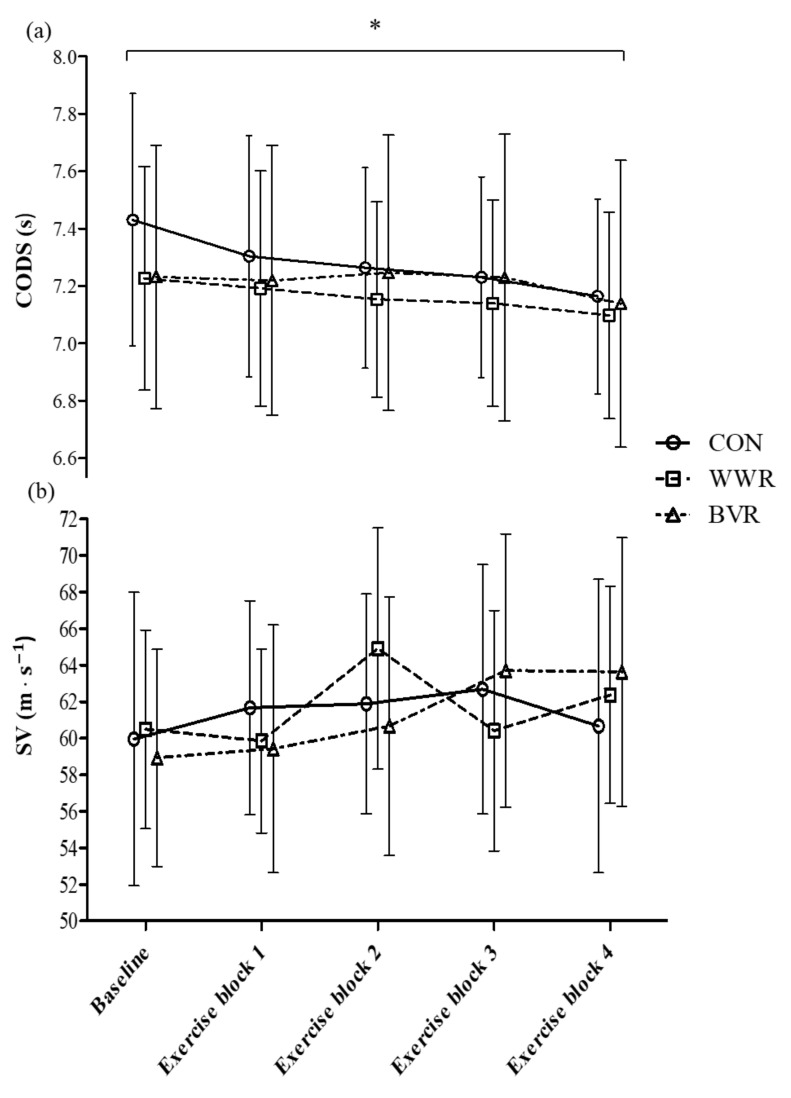
Change of direction speed (CODS) (**a**) and smash velocity (SV) (**b**) across all five timepoints. * Main effect of time *p* < 0.05, and post hoc tests revealed significantly faster CODS at exercise block 4 as compared to baseline and exercise block 1.

**Figure 4 jfmk-06-00081-f004:**
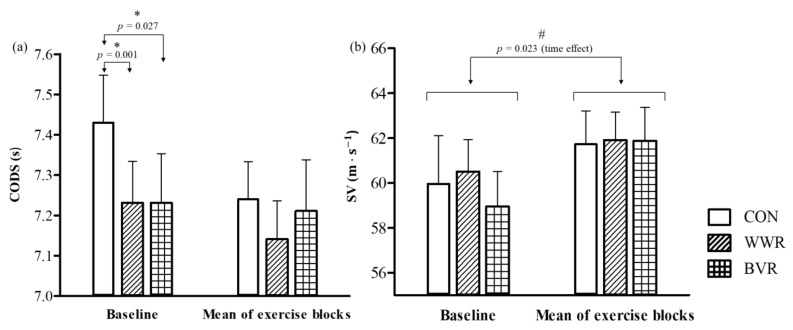
Change of direction speed, CODS (**a**) and smash velocity, SV (**b**) at baseline relative to mean values of exercise blocks 1 to 4. * Interaction effect of trial × time *p* < 0.05, and post hoc tests revealed significantly faster CODS in weighted wearable resistance (WWR) and banded variable resistance (BVR) as compared to CON condition at baseline. # Main effect of time *p* < 0.05, and post hoc tests revealed significantly faster SV during the 4 exercise blocks than at baseline under all three warm-up conditions.

**Figure 5 jfmk-06-00081-f005:**
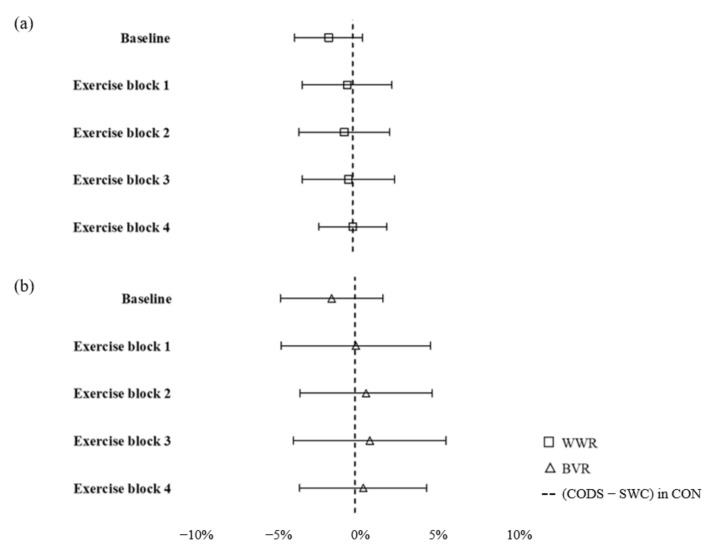
Percentage difference (%DIFF) of change of direction speed (CODS) in WWR (**a**) and BVR (**b**) vs. CODS–SWC in CON at each measured timepoint. SWC observed at baseline, exercise block 1 to exercise block 3 in (**a**) and baseline in (**b**).

**Table 1 jfmk-06-00081-t001:** Exercises involved in the 3 different warm-ups of control (CON), weighted wearable resistance (WWR), and banded variable resistance (BVR).

Phases	Exercises	Repetitions/Distance
General mobility— Exercises 1 to 6 Dynamic stretching and activation— Exercises 7 to 19 Sport-specific plyometric— Exercises 20 to 24	Jogging Backpedal Side Shuffle (alternate lead foot) Carioca (alternate lead foot) Jogging with high knees Jogging with butt kicks Walking on toes Walking on heels Walking hip circles (medial to lateral) Walking hip circles (lateral to medial) Walking deep lunges Thoracic rotation in lunge position (both sides) Frontal leg swings (both legs) Lateral leg swings (both legs) Alternate side lunges Wall angels Dynamic arm swings (forward and backward) Side arm swings Fast feet running Plyometric push-ups *# Overhead smash ^ Bilateral countermovement jumps Alternating split squat jumps 2-corners shadow movement	4 × 10 m 4 × 10 m 4 × 10 m 2 × 10 m 2 × 10 m 2 × 10 m 2 × 10 m 2 × 10 m 2 × 10 m 2 × 10 m 2 × 10 m 5 reps per leg 10 reps per leg 10 reps per leg 5 reps per leg 10 reps 10 reps 10 reps 1 × 10 m 2 × 5 reps 2 × 5 reps 5 reps 3 reps per leg 2 × 2 rounds

* If females are unable to perform traditional push-ups, they may perform push-ups with both knees on the floor. # For BVR condition, traditional push-ups were performed instead of plyometric push-ups due to the heavy resistance bands used. ^ For WWR condition, wrist weights were worn in addition to weighted vest during overhead smash.

**Table 2 jfmk-06-00081-t002:** Descriptive statistics of change of direction speed (CODS) and smash velocity (SV) performance tests of the control (CON), weighted wearable resistance (WWR), and banded variable resistance (BVR).

	Warm-Up Conditions		
	CON	WWR	BVR
Change of direction speed, CODS (s)			
Baseline	7.43 ± 0.44	7.23 ± 0.39	7.23 ± 0.46
Exercise block 1	7.30 ± 0.42	7.19 ± 0.41	7.22 ± 0.47
Exercise block 2	7.26 ± 0.35	7.15 ± 0.34	7.26 ± 0.48
Exercise block 3	7.23 ± 0.35	7.14 ± 0.36	7.23 ± 0.50
Exercise block 4	7.16 ± 0.34	7.10 ± 0.36	7.15 ± 0.50
Mean (of exercise blocks 1 to 4)	7.24 ± 0.38	7.14 ± 0.38	7.21 ± 0.52
Mean (of all five timepoints) *	7.28 ± 0.38	7.16 ± 0.37	7.21 ± 0.48
Smash velocity, SV ($m\cdot s^{-1}$)			
Baseline	60.0 ± 8.0	60.5 ± 5.4	58.9 ± 6.0
Exercise block 1	61.7 ± 5.9	59.9 ± 5.0	59.4 ± 6.8
Exercise block 2	61.9 ± 6.0	64.9 ± 6.6	60.7 ± 7.1
Exercise block 3	62.7 ± 6.8	60.4 ± 6.6	63.7 ± 7.5
Exercise block 4	60.7 ± 8.0	62.6 ± 6.0	63.6 ± 7.4
Mean (of exercise blocks 1 to 4)	61.5 ± 6.2	62.0 ± 5.3	61.6 ± 6.7
Mean (of all five timepoints) *	61.4 ± 4.0	61.6 ± 6.0	61.3 ± 7.0

* Mean of all five timepoints is the average of values at baseline and exercise blocks 1 to 4.

## Data Availability

Restrictions apply to the availability of these data. Data was obtained from participants (national badminton players) and are available from the authors with the permission of the participants.

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
