# Peer review of "Effectiveness of On-Court Resistive Warm-Ups on Change of Direction Speed and Smash Velocity during a Simulated Badminton Match Play in Well-Trained Players"

_jfmk, 2021, doi:10.3390/jfmk6040081_

Round 1

Reviewer 1 Report

Thank you to the authors for their submission to JFMK.

The study is attempts to answer a meaningful research question for athletes and practitioners working within the sport of badminton.

The study has been well designed – I particularly like that the authors have sought the examine how the PAPE effects are impacted by the sport by evaluating performance metrics at multiple time-points between the simulated match-play protocol.

I think the manuscript is extremely well written and I wish to commend the authors for their work.

Overall, I would have little hesitation in accepting the manuscript. However, I would invite the authors to consider a couple of small suggestions first.

Most notably, would it be possible to consider individual responses to the protocols?

The graphs show mean +/- SD, so this is not clear. The best way to do this would be with participant-specific SWC/MDC thresholds. However, it appears that these would not be possible as you’ve not determined reliability within this cohort. Perhaps you could consider using the group mean SD values for this, or based upon the CV values you’ve determined/referenced from the prior investigations if you deem this appropriate.

Keywords – Would suggest adding “Wearable Resistance” as a term

Introduction

Line 43 – I’m not sure about the use of ‘also’. Perhaps something like ‘in particular’?

Line 70 – Should there be a comma instead of ‘and’ between “Judo Fitness Test and peak power…”?

Line 74 – Badminton does not need to be capitalised here.

Line 75 – Change ‘of’ to ‘such as’.

Methods

Line 85-86 – Could participant characteristics be presented separately for males and females?

Could you also add personal best world or national rankings for participants? Normally attainable from something like tournamentsoftware platform.

Could you provide information regarding disciplines too? Singles/Doubles/Mixed.

Line 200 – Did you look at success rate (i.e. accuracy) of smashes? Would be interesting to see how this changes not just in response to warm-up, but also following the SMP.

Line 222 – Was this only for CODS? Why not for SV?

Also, as noted above, would be nice to consider this on an individual level.

Discussion

Line 298 – It appears that CODS was faster following the SMP blocks (as you’ve noted for SV – Line 346-348). This warrants a note at some point in the discussion.

Line 345 – Should this be ‘resistance’?

Line 348 – How do the SV values in the current study compare to other data? i.e. McErlain-Naylor et al, 2020. Do differences in data collection method matter – i.e. Stalker gun vs VICON?

Author Response

Response to Reviewer 1 Comments 

Thank you so much for your kind comments and constructive feedback. We really appreciate your time and effort in reviewing this article.

Comments

  1. Most notably, would it be possible to consider individual responses to the protocols?

The graphs show mean +/- SD, so this is not clear. The best way to do this would be with participant-specific SWC/MDC thresholds. However, it appears that these would not be possible as you’ve not determined reliability within this cohort. Perhaps you could consider using the group mean SD values for this, or based upon the CV values you’ve determined/referenced from the prior investigations if you deem this appropriate.

reply: Thank you for your suggestion. Yes, unfortunately we did not determine reliability in this study for both SV and CODS. For SV, we actually ran some statistical tests but there were no differences in the groups. The idea of using group mean SD values is definitely one we can consider in the future, but it would not be effective for this study since the SD is derived from both genders and the value is currently quite large; this leads to the same outcome.  

  1. Keywords – Would suggest adding “Wearable Resistance” as a term

reply: Have added into line 35

  1. Introduction

Line 43 – I’m not sure about the use of ‘also’. Perhaps something like ‘in particular’?

reply: Included “Likewise” in line 45 rather than the use of ‘also’.

Line 70 – Should there be a comma instead of ‘and’ between “Judo Fitness Test and peak power…”?

reply: Use of ‘and’ is correct, 2 separate components of Judo Fitness Test and peak power of barbell high pull. – line 73

Line 74 – Badminton does not need to be capitalised here.

reply: Noted, have edited it – line 77

Line 75 – Change ‘of’ to ‘such as’.

reply: Noted, have changed it – line 78

  1. Methods

Line 85-86 – Could participant characteristics be presented separately for males and females?

reply: Yes, have calculated and presented separately - line 88 - 91

Could you also add personal best world or national rankings for participants? Normally attainable from something like tournaments oftware platform.

reply: Most of the players tested are relatively young ~19+ years of age, and in their midst of transition to the professional circuit. Majority of them do not have a substantial BWF world ranking at the present. However, the players chosen for this study were ranked at least top 8 nationally – line 93 & 95.

Could you provide information regarding disciplines too? Singles/Doubles/Mixed.

reply: Included in line 89-91

Line 200 – Did you look at success rate (i.e. accuracy) of smashes? Would be interesting to see how this changes not just in response to warm-up, but also following the SMP.

reply: Unfortunately, we did not look at the accuracy of the smashes in this study as our focus was on the velocity of the explosive smashes. The instruction for the players was to put in maximal effort in all their smashes – line 205.

Line 222 – Was this only for CODS? Why not for SV?

reply: Yes, the investigators have also discussed on this previously and noted that there were trends in the differences between the conditions for CODS but not for SV.

Also, as noted above, would be nice to consider this on an individual level.

reply: As above.

  1. Discussion

Line 298 – It appears that CODS was faster following the SMP blocks (as you’ve noted for SV – Line 346-348). This warrants a note at some point in the discussion.

Reply: Included discussion point on line 386 onwards.

Line 345 – Should this be ‘resistance’?

Reply: We have checked, vertical ground reaction force is correct.

Line 348 – How do the SV values in the current study compare to other data? i.e. McErlain-Naylor et al, 2020. Do differences in data collection method matter – i.e. Stalker gun vs VICON?

reply: The SV values in our study compare well with the study of Phomsoupha & Laffaye et al., 2014., whose protocol we took reference from. The difference in data collection method matters - as the point of the shuttlecock trajectory which the stalker gun (fixed in position) measures is different for each individual player, the shuttlecock velocity will likewise vary. The VICON could measure the shuttlecock velocity directly after the smash due to available 3D analysis. However, we do not have access to the VICON system for our study.

Reviewer 2 Report

The manuscript is well written, the topic is of scientific interest and the outcomes may find potential practical application in badminton performance since it is highlighted that badminton players can gain a competitive edge against their rivals at least at the start of the match play by including resistance of approximately 10% of their body weight to their plyometric warm up exercises. 

Specific Comments

  1. At what period of the season did the measurements take place (i.e. off-season, pre-season, in-season)? Please, note it.
  2. Consider presenting some main body composition data of the players since you used an Inbody 770 body composition analyser as you mentioned in methods section.
  3. Were all the measurements conducted by the same investigator? This should be noticed.
  4. The size of the sample is rather small and particularly the number of females.

Author Response

Response to Reviewer 2 Comments 

Thank you so much for your kind comments and constructive feedback. We really appreciate your time and effort in reviewing this article.

Comments

  1. At what period of the season did the measurements take place (i.e. off-season, pre-season, in-season)? Please, note it.

reply: Off-season phase –  we have included it in line 104-105

  1. Consider presenting some main body composition data of the players since you used an Inbody 770 body composition analyser as you mentioned in methods section.

reply: The investigators did consider presenting the main body composition data but after much discussion, decided to just input the necessary data of limb mass and body mass as the accuracy of the Inbody 770 may not be the best.

  1. Were all the measurements conducted by the same investigator? This should be noticed.

reply: thanks you, we have included this in line 105-106

  1. The size of the sample is rather small and particularly the number of females.

reply: Yes, as mentioned, in line 91 onwards, “Stringent inclusion criteria were adhered to in the recruitment of participants as performance tests required high skill levels - players must be nationally ranked either in the singles or doubles’ categories, with at least two years of experience with full-time strength and conditioning training (minimally two sessions a week).” It was also unfortunate that a few of the players were nursing injuries. That limited the number of participants we could recruit, especially on the female side.